# Cooperativity and Interdependency between RNA Structure and RNA–RNA Interactions

**DOI:** 10.3390/ncrna7040081

**Published:** 2021-12-15

**Authors:** Ilias Skeparnias, Jinwei Zhang

**Affiliations:** Laboratory of Molecular Biology, National Institute of Diabetes and Digestive and Kidney Diseases, Bethesda, MD 20892, USA; ilias.skeparnias2@nih.gov

**Keywords:** RNA, RNA structure, RNA–RNA interactions, base pair, base stacking, riboswitch, ribozyme, tRNA, T-box, RNase P

## Abstract

Complex RNA–RNA interactions are increasingly known to play key roles in numerous biological processes from gene expression control to ribonucleoprotein granule formation. By contrast, the nature of these interactions and characteristics of their interfaces, especially those that involve partially or wholly structured RNAs, remain elusive. Herein, we discuss different modalities of RNA–RNA interactions with an emphasis on those that depend on secondary, tertiary, or quaternary structure. We dissect recently structurally elucidated RNA–RNA complexes including RNA triplexes, riboswitches, ribozymes, and reverse transcription complexes. These analyses highlight a reciprocal relationship that intimately links RNA structure formation with RNA–RNA interactions. The interactions not only shape and sculpt RNA structures but also are enabled and modulated by the structures they create. Understanding this two-way relationship between RNA structure and interactions provides mechanistic insights into the expanding repertoire of noncoding RNA functions, and may inform the design of novel therapeutics that target RNA structures or interactions.

## 1. Introduction

Biomolecules have evolved to function in coordination with each other in crowded intracellular environments in which new molecular interfaces and connections develop continuously. Direct protein–protein interactions likely mediate the bulk of cellular communication and signaling pathways in contemporary organisms. In the primordial RNA world, RNA–RNA and RNA–metabolite interactions were responsible for executing and regulating most if not all aspects of the metabolism. Not all of these RNA interfaces have been lost. RNA–metabolite interactions remain key pathways that regulate bacterial metabolism in the form of diverse riboswitches [1]. RNA–RNA interactions mediate numerous cellular processes to this day, including RNA processing, maturation and modifications (by the spliceosome, RNase P, snoRNP, etc.), gene expression regulation (by sRNAs, T-boxes, miRNAs, siRNAs, piRNAs, etc.), mRNA translation (by ribosomes, IRESes), viral replication and antiviral immunity, etc. There are even newly acquired structural and regulatory elements that operate via RNA–RNA interactions, such as some mammalian-specific long noncoding RNAs (lncRNAs) [2,3].

In contrast to our more detailed understanding of protein–protein interfaces and their specificities, the physical picture is much less clear for how RNA–RNA interactions occur. Even for relatively simple RNA–RNA interactions that are primarily driven by base pairing, it is generally not clear how one RNA finds its target RNA partner. Is either RNA partially structured prior to and during the encounter? The vast majority of RNA–RNA interactions are multivalent, as they involve more than one base pair and frequently more than one interface. In what order are these individual contacts established, how do they propagate, and which among the multiple contacts determine the specificity and affinity?

To address these fundamental questions about the origin and characteristics of RNA–RNA interactions such as affinity and specificity, we first explore the chemical basis and underlying forces that drive RNA–RNA contacts. We then examine three primary classes of RNA–RNA interactions—those between single-stranded RNAs (ssRNAs), between ssRNA and structured RNAs, and between structured RNAs. Using specific examples from triplexes to ribozymes, with particular emphasis on those with high-resolution structural information, we derive general trends and themes that are common to these diverse systems. These insights could help understand other RNA–RNA interaction systems whose structures remain unknown, and further inform the design of strategies or therapeutics that modulate RNA–RNA interactions important for cellular physiology or pathology.

## 2. Different Modalities of RNA–RNA Interactions: Components and Underlying Forces

The diverse ways RNAs can interact with each other are determined by the chemical, electrostatic, and geometric properties of their three components: the phosphodiester linkage, the ribose with its characteristic 2′-hydroxyl group, and the planar, heterocyclic nucleobase. While the latter, ribose- and nucleobase-mediated contacts are the primary drivers of RNA–RNA interactions, long-range interactions that involve the phosphodiester backbone are surprisingly common [4]. These generally involve OP1 or OP2 atoms as hydrogen bond acceptors and 2′-hydroxyl groups of an adjacently packed RNA segment as donors, forming the “ribose-phosphate zipper” motif [4]. It is possible that strong cationic ions such as Mg^2+^ or polyamines are needed to partially shield the charge of the phosphate backbone to allow such juxtaposition. Both the overall trajectory and local structure of the RNA phosphate backbone are more frequently recognized by basic amino acid residues of sequence-non-specific RNA-binding proteins. The ribose 2′-hydroxyl imparts significant structural rigidity over DNA strands and enforces A-form geometry in both double-stranded RNAs (dsRNA) and RNA–DNA hybrids. It also plays key roles in RNA recognition and ribozyme catalysis [5]. The versatile ribose 2′-hydroxyls can act as both donors and acceptors of hydrogen bonds, thereby allowing them to pair up and stitch together the backbone of adjacent antiparallel strands. These 2′-hydroxyl-2′-hydroxyl linkages, buttressed by neighboring 2′-hydroxyl contacts with purine N3 or pyrimidine O2 groups, form the prevalent “ribose zipper” motif, which contribute about ~2 kcal/mol of stabilizing energy [6,7].

The planar, aromatic nucleobases are the principal features that drive specific RNA–RNA interactions and can interact among themselves or with ribose 2′-hydroxyls and bridging and non-bridging oxygens of the phosphate backbone. Nucleobases can base pair or stack with other nucleobases and pack with ribose sugars via sugar-π interactions. Each nucleobase contains three geometric edges with distinct chemical compositions, namely the Watson–Crick, Hoogsteen, and sugar edges [8]. Nucleobases can further assume the *anti* (common) or *syn* (rare) configuration via the rotation of the glycosidic bond. Each edge can form base-pairing interactions with the same or a different edge *in trans* or *in cis*, thus producing a large variety of possible pairing configurations, most of which have been experimentally observed [8]. In addition to pairing interactions, whose vectors lie near perpendicular to the backbone trajectory, the aromatic nucleobases can stack along their backbone vector via π-π interactions. Importantly, if base-pairing interactions primarily create the local structures of RNA, stacking interactions principally control the global shape and trajectory of the RNA. This is exemplified by the double-helix shape of duplex nucleic acids, whose characteristic shape is largely a result of concatenated base stacking in the parallel displaced configuration rather than by base pairing. In addition, base stacking frequently mediates RNA–RNA interactions. Base intercalation is a key interaction that mediates the formation of the canonical tRNA elbow structure, where the G18 residue of D-loop inserts into the stacking gap of the T-loop [9,10,11], as well as the interdigitated double T-loop motif (IDTM) found in most T-box riboswitches and RNases P, which is formed by the reciprocal intercalation of two T-loops in head-to-tail opposition [12,13].

## 3. RNA Strand Assembly: From ssRNA to dsRNA, Triplex to G-Quadruplexes

All RNA transcripts start as ssRNA strands that still inherently repel each other from their negative charges. In this form, a balancing act between intra-strand stacking and repulsion from the neighboring phosphates controls the conformation of the ssRNA. As a result, polyA chains assume helical stacks due to strong stacking, while poorly stacked polyU chains form self-avoiding flexible polymers. The substantial hydrophobicity of the nucleobases motivates their approach towards each other against the electrostatic repulsion, which during their brief encounters explore ways to form base pairs and base stacks. For complementary ssRNA strands, once a few seed pairs form in the expected Watson–Crick geometry, the pairing propagates to lengthen the duplex in both directions. This pairing creates the basic building block of RNA secondary structure, in the form of a linear, antiparallel duplex.

Once formed, individual dsRNA segments are strongly inclined to stack coaxially with each other to form longer assemblies or “spines”. These end-to-end interactions are driven by favorable enthalpies from the π–π stacking between their termini base pairs, as well as favorable entropies associated with burying hydrophobic nucleobases that are otherwise exposed to the solvent. Such spontaneous coaxial assembly of dsRNA helices is exemplified by the tRNA folding process, during which the four helical segments stack coaxially into two longer stacks, before being joined at the elbow [14,15,16]. In the majority of RNA crystals, the helical segments are seen to form pseudo-infinite helices via end-to-end, coaxial stacking [17].

RNA duplexes generally exhibit A-form geometry characterized by deep, narrow major grooves and wide, shallow minor grooves. Both grooves can interact with an incoming ssRNA strand, sometimes at the same time with the same ssRNA, to form RNA triplexes. Notably, major groove triplexes that involve a third “Hoogsteen strand” are more stable than their minor groove counterparts [18], presumably due to the larger Hoogsteen edge in the major groove available for base triple formation than the sugar edge in the minor groove. First discovered by Gary Felsenfeld, David Davies, and Alexander Rich in 1957 in vitro [19], naturally occurring RNA triplexes have since then been identified in the telomerase RNA, spliceosome, Group II intron ribozymes, various riboswitches, and RNA stability elements such as the element for nuclear expression (ENE) [18]. Triplexes are also frequently found in RNA pseudoknots [20].

Beyond three-stranded RNAs, four strands of G-rich ssRNA can also interact among themselves to form G-quadruplexes. These extensively paired, highly stacked structures exhibit diverse topological arrangements, are more stable in RNA compared to DNA, and tend to fold slightly differently, favoring the parallel alignment in RNA form [21,22]. Serendipitously discovered by Martin Gellert, Marie N. Lipsett, and David R. Davies in 1962 [23], these robust and pervasive structures are stabilized by strong stacking interactions between adjacent G-quartet planes, which are quadrangular base pairs formed via tandem, circularizing interactions between their Watson–Crick and Hoogsteen edges. Interestingly, the large, hydrophobic quartet surfaces, if exposed to the solvent, can mediate robust intermolecular stacking interactions, as seen in the case of the dimeric, fluorogenic Corn RNA in the presence or absence of its chromophore ligand 3,5-difluoro-4-hydroxybenzylidene imidazolinone-2-oxime (DFHO) [24].

In summary, RNA–RNA interactions occur through diverse types of contacts (base pairing, base stacking, ribose zipper, A-minor, sugar-π interactions, etc.) and basic strand configurations (ssRNA, dsRNA, triplexes, and G-quadruplexes, etc.) More detailed analyses of the chemical structures of major RNA structural motifs such as K-turns, T-loops, A-minor motifs, and more, have been reviewed previously [25,26,27,28,29]. Next, we will classify RNA–RNA interactions into three archetypes: those between ssRNAs, between ssRNA and structured RNAs, and between structured RNAs.

## 4. Interaction between ssRNAs: Base Pairing and Beyond

The most common type of RNA–RNA interactions are base-pairing interactions between complementary ssRNA strands. Prokaryotic small regulatory RNAs (sRNAs) target mRNAs *in cis* or *in trans*, primarily to repress gene expression by occluding the ribosome-binding sites, and secondarily by inducing RNase E cleavage via the dsRNA structure [30,31]. The establishment of such sRNA–mRNA pairing interactions frequently requires the action of RNA-binding chaperone proteins such as Hfq, ProQ, FinO, etc. (Figure 1a) [32,33]. Taking this protein-chaperoned RNA–RNA pairing one step further, the targeting ssRNA can first assemble into effector ribonucleoprotein (RNP) complexes before finding and annealing with the target mRNA. This mode of operation is exemplified by several RNA-targeted Clustered Regularly Interspaced Short Palindromic Repeats (CRISPR) systems including Type II (Cas9), Type III (Csm/Cmr), and Type VI (Cas13) CRISPRs [34,35]. Other types of sRNA may have more complex mechanisms of action, such as the 514-nt Staphylococcus aureus RNAIII. RNAIII, one of the largest sRNA identified so far, is proposed to repress the translation initiation of multiple mRNAs by annealing its CU-rich ssRNA loops with the complementary Shine–Dalgarno sequences of the mRNA [36,37,38].

In eukaryotes, there are at least three major categories of short ssRNAs that function analogously to the prokaryotic sRNAs and CRISPR RNA (crRNAs): the small interfering RNAs (siRNA), microRNAs (miRNA), and PIWI-interacting RNAs (piRNA) (Figure 1b,c). Each of these ssRNAs is processed from RNA hairpins or longer ssRNAs, and subsequently handed over to effector Argonaute (AGO) proteins, forming RNA-induced silencing complexes (RISC) of different flavors (siRISC, miRISC, and piRISC) [39,40,41]. These RNP complexes then scan for their target mRNAs and once found either catalyze their cleavage or repress their translation. Interestingly, at least with miRISC, the ~22-nt-long miRNA can sequentially form two discontinuous segments of base-pairing, first with the seed region (nt 2–7), and subsequently with the supplementary pairing region (nt 13–16). The latter is enabled by a conformational change triggered by the initial seed–mRNA pairing. Structural analyses revealed that the seed–target duplex minor groove is inspected by AGO to ensure Watson–Crick pairing and reject altered pairing geometry such as G•U wobble pairing [42].

A notable variation in ssRNA–ssRNA interactions occurs when both ssRNAs are hosted in structured hairpin stem loops. Such interactions can mediate long-range “kissing” interactions that bring far-flung RNA helices together to form compact structures (Figure 1d). Perhaps the most notable intermolecular ssRNA–ssRNA interactions aided by RNA structure are the codon–anticodon interactions between the mRNA and the tRNAs on the ribosome [43]. This interaction is driven by partial structure formation of both RNA partners. The mRNA bound by the ribosome assumes an extended conformation poised for the incoming tRNAs, which has evolved a particular anticodon stem loop structure that pre-stacks the anticodon trinucleotide in a helical ssRNA trajectory [43]. The short 3-bp mRNA–tRNA duplex is axially stabilized by adjacent tRNA and rRNA contacts, such as cross-strand stacking by the R37 residue of the tRNA on one side and the C1400 residue of the 16S rRNA (*Escherichia coli* numbering) on the other [44,45,46]. The codon–anticodon duplex is further laterally stabilized, and its Watson–Crick geometry enforced by the minor group interactions from G530, A1492, and A1493 of the rRNA [44,45,46]. Intriguingly, similar base-pairing, helix-capping by cross-strand stacking, and supplementary minor groove contacts also occur outside the ribosome between the tRNA and the T-box riboswitches [13,47,48].

## 5. Interaction between ssRNA and Structured RNAs

In addition to ssRNAs interacting among themselves, ssRNA can also bind to complementary ssRNA regions present on structured RNAs. Such interactions, when occurring *in cis* in the same RNA, frequently lead to pseudoknot formation. When occurring *in trans*, such interactions allow the structured RNAs to store or titrate the ssRNA, or enable the ssRNA to modulate the conformation or stability of the larger structured RNA (Figure 2a). In the former case, some circular lncRNAs are proposed to act as miRNA sponges, by serving as reservoirs to store miRNAs complementary in sequence, and conditionally release them subsequently [49]. In the latter case, ssRNA can interact with the structured RNA in at least three different ways that lead to diverse outcomes.

First, when an ssRNA encounters a dsRNA hairpin composed of a stem and a distal loop or side bulge, which is partially complementary to the ssRNA, they could base pair to form a new helical segment. If the newly paired region extends to either the 5′ or 3′ edge of the loop, the newly formed intermolecular stem would be in a position to spontaneously stack coaxially with the existing stem of the hairpin. This can substantially stabilize the hairpin structure, as illustrated by the functionally important tRNA 3′-end interaction with the T-box riboswitch antiterminator. When the tRNA 3′-NCCA terminus forms four base pairs with a complementary NGGU tetranucleotide located at the 5′ edge of the antiterminator bulge, the resulting 4-bp tRNA–T-box intermolecular duplex stacks with and, thus, stabilizes the lower A1 helix at the base of the antiterminator [50,51]. This ssRNA-stabilization of the adjacent dsRNA is key to allowing an uncharged tRNA to effect structural switching of an mRNA. Alternatively, if the ssRNA is complementary to the stem rather than the loop region, it can strand invade the duplex, destabilizing it to form a mutually exclusive helix. This simple competition and ambiguity in the pairing scheme provides a mechanistic basis for most bistable gene-regulatory RNAs such as transcription attenuators and riboswitches. A number of artificial antisense oligonucleotides (ASOs) have also been designed based on this mechanism, where an exogenous oligo can perturb RNA structure to modulate mRNA splicing, IRES (internal ribosome entry site) structure and mRNA translation, mRNA stability, etc. In a third scenario, an ssRNA that is not complementary to either the loop or the stem can bind the dsRNA in its major or minor groove, or both, to form RNA triplexes [18]. The best-characterized example is the widespread expression and nuclear retention elements (ENEs) found in the 3′ regions of several genomic lncRNAs (such as MALAT1 and MENβ) and viruses (such as Kaposi’s sarcoma-associated herpesvirus or KSHV), which protect the 3′ end of the RNA from deadenylation and decay [52,53,54].

Comparing these three scenarios, the binding of the incoming ssRNA would generally stabilize the original RNA structure if the binding is to the loop region, and destabilize if the ssRNA invades the dsRNA stem, and stabilize again if the ssRNA binds forming triplexes. Recent structural and modeling studies have shed new light on the formation and function of triplexes, as discussed below for the ENE and the CCR5 examples.

### 5.1. ENE Triplexes for RNA Stability

The ENE elements have been identified in more than 200 host and viral genes near the RNA 3′ ends, which shield the 3′ polyA or the 3′ end from deadenylases and RNases, thereby stabilizing their upstream RNA (Figure 2b) [18,54]. The ENE is a hairpin structure that contains a long internal loop in the middle section, both strands of which primarily contain uridines. The U-rich internal loop is poised to receive an ssRNA polyA sequence, which will pair with the antiparallel U-rich strand forming dsRNA. The other U-rich strand of the ENE, parallel to the polyA, will then occupy the major groove of the newly formed dsRNA, forming a major groove triplex. Other adjacent contacts such as tandem A-minor interactions and coaxial stacking further stabilize the U•A-U triplex. In vitro, the MALAT-1 ENE element can robustly associate with a polyA oligo in trans with a *K*_d_ of 20 nM forming the triplex [54]. Since the polyA oligo is a fairly rigid ssRNA stack, this interaction initially occurs between the flexible, unpaired polyU internal loop and the incoming polyA stack, and gradually takes shape. An alternative scenario is that a flexible ssRNA can associate with preformed dsRNA structure forming a triplex, as exemplified by the CCR5 triplex [55].

### 5.2. CCR5 Triplexes for Ribosome Frameshifting Regulation

The mRNA of the human CCR5 gene, a coreceptor of HIV-1 entry into T-cells, contains a structured programmed −1 ribosomal frameshift (−1 PRF) element (Figure 2c) [55,56]. The CCR5 −1 PRF folds into a pseudoknot structure and impedes the translating ribosome. About 15% of the paused ribosomes were shifted one nucleotide upstream relative to the mRNA into the −1 reading frame, directed to a premature termination codon, and decayed through the nonsense-mediated decay (NMD) pathway. Interestingly, host miRNAs regulate the stability of the CCR5 pseudoknot by direct dissociation. Computational modeling, MD simulation, and in vitro analyses suggest that miR-1224 binds the dynamic pseudoknot with sub-nanomolar affinity, along the extended minor groove of its dsRNA region [55]. SHAPE (Selective 2’ Hydroxyl Acylation analyzed by Primer Extension) analysis detected no major conformational changes in the pseudoknot, suggesting that the miRNA binds to a preformed structure and reinforces it, presumably to counteract the robust helicase activity of the incoming ribosome. Compared to the ENE triplexes, the CCR5 triplex belongs to the minor groove class that is thought to be less stable than their major groove counterparts, possibly due to the more expansive Hoogsteen edge in the major groove than the sugar edge in the minor groove. Its remarkable binding affinity hints that there are likely to be substantial tertiary contacts in addition to the mRNA–miRNA base pairing. A full understanding of these interactions awaits determination of its experimental structure. Interestingly, another −1 ribosome frameshifting pseudoknot element from beet western yellow virus (BWYV) contains a minor groove triplex that could share structural similarities with the CCR5 triplex [57]. In this triplex, an adenine ladder from the loop 2 binds in the minor groove of stem 1 forming a triplex via 4 nearly contiguous A-minor interactions (Figure 2e).

Another paradigm of ssRNA interaction with structured RNA is the small nucleolar RNAs (snoRNAs) (Figure 2d). SnoRNA guides recognize their target segments in the rRNA via base-pairing and potentially other contacts to introduce 2′-O-methyl (Box C/D) or pseudouridine (Box H/ACA) modifications that are important for ribosome biogenesis and function [58,59,60]. The details of the RNA–RNA and RNA–protein contacts in the eukaryotic snoRNPs remain unclear and await structural analyses of the full complexes [60,61,62].

## 6. Higher-Order Interactions between Structured RNAs

Partially or wholly structured RNAs can also recognize each other to mediate a number of cellular transactions from gene regulation to RNA processing and modification. These include localized interactions between compact RNA structural motifs as well as larger, multivalent interactions that involve several discreet interfaces [63,64]. Certain RNA motifs have evolved to form stable structural folds, which can then be recognized by their cognate partner motifs [29]. This is exemplified by the complementary GNRA tetraloop and tetraloop–receptor motifs, which act as specific fasteners that allow RNA helices to dock with each other [65,66], and the T-loop–D-loop interaction that generates the tRNA elbow [14]. Besides these asymmetric lock and key contacts, some structural motifs have structural features that allow them to interact among themselves to make symmetric or pseudo-symmetric dimers, as seen in how two T-loop motifs intercalate and pair with each other, forming the Interdigitated Double T-loop Motif (IDTM) [13,48,67].

Larger RNAs that are equipped with multiple such interaction motifs are capable of concurrent interactions, enabling them to make multivalent interactions with other structured RNAs. An artificial example is a designed RNA helix that contains both the tetraloop and tetraloop receptor, spaced so that two such RNAs can bind head-to-tail to form dimers with a robust *K*_d_ of ~5 nm [68,69]. Among structurally elucidated natural complexes between structured RNAs, most contain the tRNA as a partner or macromolecular ligand. This fact is unsurprising, as cellular tRNAs are among the most abundant, structurally defined, and solvent-accessible RNAs that traverse multiple subcellular boundaries and compartments (nucleus, cytoplasm, mitochondria, etc.) and transit between numerous cellular machineries from the aminoacyl-tRNA synthetases (aaRSes) that aminoacylate the tRNAs to the ribosomes that consume the amino acid cargo and eject the deacylated tRNAs [70,71,72,73].

Below, we discuss the RNA–RNA interactions that underlie known examples of such complexes, of tRNAs in complex with the HIV-1 primer-binding site (PBS), RNase P, the T-box riboswitches, and the SAM-Ixcc riboswitch.

### 6.1. HIV-1 PBS-Host tRNA Interaction

The constant exposure of host tRNAs to viral proteins and RNAs that populate the cytosol have provided ample opportunities for tRNAs to be manipulated and exploited for viral gains. Indeed, most retroviruses such as HIV-1 all rely on host tRNAs as obligatory reverse transcription primers [74]. tRNAs are further tapped by HIV-1 as regulators of Gag polyprotein localization to the plasma membrane through its N-terminal matrix domain, to optimize the timing of virion assembly [75,76,77,78,79,80]. In order to initiate the replication of its RNA genome, HIV-1 hijacks the host tRNA^Lys3^, melts its 3′ 18-nt region and anneals it with a complementary region on the HIV-1 genomic RNA termed the PBS (primer-binding site) (Figure 3a) [81,82,83]. HIV-1 reverse transcriptase (RT) subsequently forms a ternary complex with the HIV RNA–tRNA complex to initiate reverse transcription from the tRNA 3′-end [84]. A recent structure of this ternary complex visualized a substantial structural rearrangement of the tRNA^Lys3^ primer [85], in which the tRNA refolds into a single extended hairpin by melting and re-annealing of its 5′-terminal strand with the 5′ strand of the T stem. This strand migration helps displace and liberate the 18-nt-long, 3′-terminal anti-PBS element of the tRNA, and allows it to pair with the PBS. This complex is further stabilized by multi-segment coaxial stacking between three discreet dsRNA elements, which may also help offset the initial energetic cost of deforming the tRNA [85].

### 6.2. RNase P-pre-tRNA Interaction

A series of processing steps and the well-orchestrated action of essential ribonucleases are required for the maturation of tRNAs and rRNAs. Among those, the ribonuclease P (RNase P) is responsible for removing the 5′ leader sequence from the precursor tRNA (pre-tRNA) [86,87,88]. RNase P is present in all three kingdoms of life and exhibits a remarkable expansion of its subunit composition, similar to that of the RNA polymerase and ribosomes, which scale with the increased organism complexity [89,90]. Bacterial RNase P consists of a single RNA subunit and a sole C5 protein, while the archaeal holoenzyme contains one RNA and four protein subunits [91,92]. By contrast, human RNase P consists of a single catalytic RNA (RPPH1, Ribonuclease P RNA Component H1) and ten protein subunits (Rpp14, Rpp20, Rpp21, Rpp25, Rpp29, Rpp30, Rpp38, Rpp40, Pop1, and Pop5) [93]. The increased number of accessory proteins and their growing impact on the core ribozyme activity may reflect fine-tuning of its catalytic properties, expansion of its substrate repertoire beyond pre-tRNAs, or the evolution of additional regulatory interfaces with other cellular machineries. Interestingly, in addition to the maturation of pre-tRNAs and rRNAs, RNase P also processes and matures the 4.5S rRNA of the signal recognition particle (SRP) in bacteria and major lncRNAs such as MALAT1 and MEN-β/NEAT1 in higher eukaryotes [94,95,96]. Further, recent studies have shown that several protein subunits of the RNase P holoenzyme perform additional tasks beyond RNA processing such as regulation of chromatin organization and function, DNA replication, transcription, and translation [86].

**Figure 3 ncrna-07-00081-f003:**
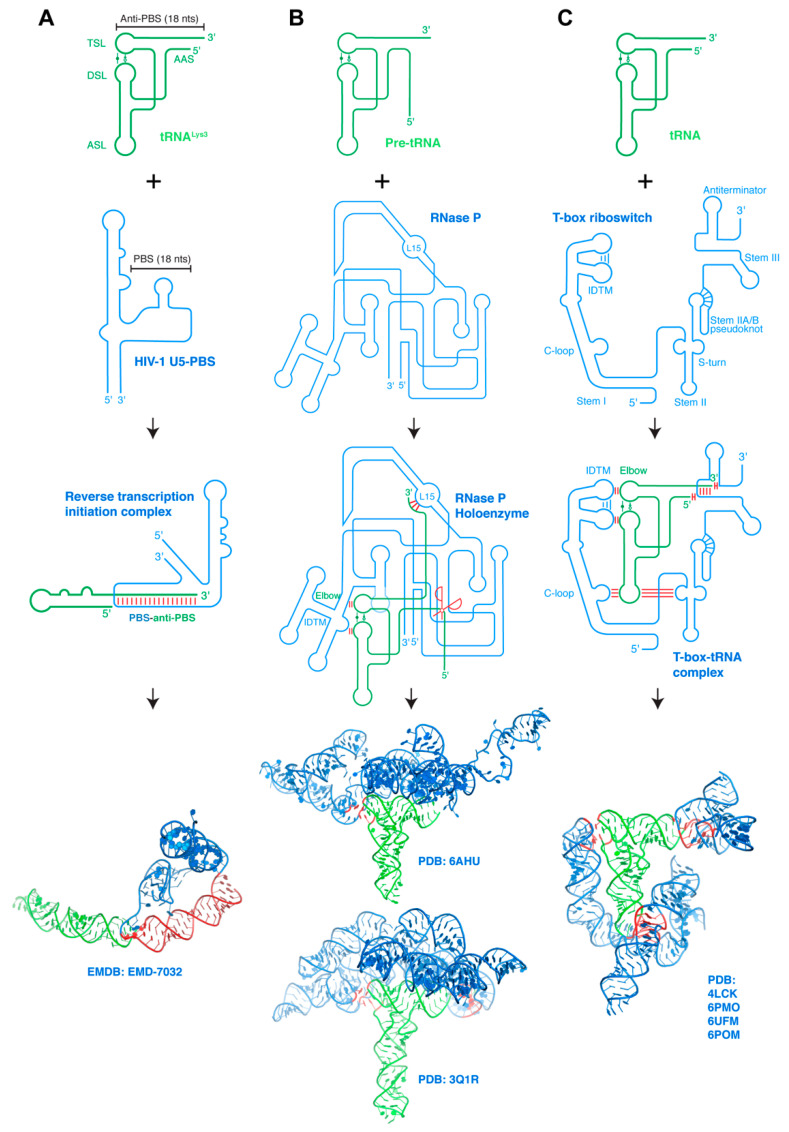
Interaction between structured RNAs. (**A**) Host tRNA^Lys3^ refolds and binds HIV-1 PBS genomic RNA region forming the reverse transcription initiation complex (RTIC). HIV-1 Reverse transcriptase (RT) bound to the RNA–RNA complex is omitted for clarity. Bottom: cryo-EM structure of the RTIC. tRNA is in green whereas the partner RNA is in blue. Interaction regions are highlighted in red. (**B**) Pre-tRNAs bind RNase P ribozymes through three base-pairing interactions to the tRNA 3′ region and stacking interactions with its elbow. Ribozyme catalytic center is indicated by the red scissors. Bottom: cryo-EM structure of human RNase P holoenzyme (upper) [93] and co-crystal structure of *Thermotoga maritima* RNase P holoenzyme (lower) [12]. (**C**) T-box riboswitches encase their cognate tRNA ligands via coordinated contacts at three distant interfaces: codon–anticodon base-pairing interactions stabilized by cross-stranding stacking and A-minor interactions with the Stem II S-turn, stacking interactions with the tRNA elbow, and both pairing and stacking interactions with the tRNA termini. Bottom: composite structural model of a full-length, feature-complete T-box riboswitch based on several co-crystal and cryo-EM structures [97]. Intermolecular base-pairing interactions are indicated by red sticks perpendicular to the RNA strands, intermolecular stacking interactions by two red sticks parallel to the RNA strands. RNA–RNA interfaces on structural renderings are highlighted in red.

The RNA component of RNase P recognizes the pre-tRNA through base-pairing interactions to its 3′-region and stacking interactions with its flat, surfaced-exposed elbow structure (Figure 3b) [12,14,93]. This achieves a high degree of shape complementarity with the upper half of the pre-tRNA, and allows RNase P to process all pre-tRNAs that carry diverse anticodon sequences and different variable stems. Specifically, the co-crystal structure of the *Thermotoga maritima* RNase P holoenzyme revealed that the characteristically flat elbow structure of pre-tRNA^Phe^ is recognized by stacking with two conserved regions (CR) termed CR-II and CR-III, each forming a pentanucleotide T-loop motif [98] and interdigitates with each other to form the IDTM [13,48,99,100]. In addition, the L15 loop of the catalytic domain forms three Watson–Crick base pairs with the 3′ region of the tRNA, which are in turn stabilized by metal ions and a ribose zipper [12,101,102,103]. The recent cryo-EM structure of human RNase P holoenzyme revealed that the RPPH1 RNA subunit interacts with all the protein components, which form a right hand-shaped assembly that holds the tRNA. As does bacterial RNase P, the elbow of the bound tRNA^Val^ is recognized by the CR-II and CR-III domains of the RPPH1 RNA, which interact to form the IDTM platform. Indeed, RNase Ps from different organisms, both the ribozymes and the proteinaceous RNase Ps have converged on a common ruler-like mechanism for sequence-independent pre-tRNA recognition from the elbow to the 5′-end [104]. Given the relatively simple and common RNA structure recognized by RNase P, it is all but certain that additional RNA substrates would be identified. One curious example is the MALAT1-associated small cytoplasmic RNA (mascRNA), which is located near the 3′ end of MALAT-1 lncRNAs and recruits RNase P for MALAT-1 maturation [95]. Although largely tRNA-like in its secondary structure, the mascRNA lacks the full features of the tRNA and it remains unknown how RNase P or other cellular proteins recognize this putative tRNA mimic [95,105,106].

### 6.3. T-box Riboswitch-tRNA Interactions

Similar to the RNase P ribozymes, the T-box system is another major class of structured riboregulators that directly bind tRNAs (Figure 3c) [44]. Unlike RNase P, which recognizes all pre-tRNAs irrespective of their anticodon sequence or amino acid specificity, the T-box is specific for one or several anticodons of the tRNA, and further distinguishes between aminoacylated and non-aminoacylated (or uncharged) tRNAs [107]. This dual recognition of tRNA acceptor specificity and aminoacylation state allows the T-box to monitor the individual pool of ready-to-use amino acids (in the form of aminoacyl-tRNAs), and to respond to the intracellular limitation of specific amino acids to ensure protein translation.

The T-box system was among the first RNA-centric gene-regulatory mechanisms discovered in bacteria almost three decades ago [107]. Due to the general similarity in its mechanism of gene regulation to small-molecule sensing RNAs, the T-box system has been retrospectively referred to as a riboswitch [107,108,109,110]. Nonetheless, the T-box riboswitch differs from the metabolite-binding riboswitches in recognizing a complex, structured macromolecular ligand (tRNAs), rather than a small-molecule metabolite [111] Metabolite-binding riboswitches typically fold hierarchically and cotranscriptionally, forming a local binding pocket that recognizes specific chemical groups of the metabolite [112,113]. By contrast, the sensory and gene-regulatory mechanisms of the T-box riboswitch are anchored by a set of coordinated, multivalent tRNA–mRNA interactions [48,51,97,114,115,116,117]. Since tRNA is inherently a dynamic, flexible molecule—required for its acrobatics transiting the ribosome [118]—T-boxes have evolved adaptive, malleable structural features that maintain tRNA interactions while not sacrificing selectivity [48].

T-box riboswitches share a common core sensory and regulatory mechanism [119,120]. They selectively bind to cognate tRNAs, assess their aminoacylation status to detect starvation, and regulate not only the expression of mRNAs of aminoacyl-tRNA synthetases (aaRSs) but also of numerous other proteins in other aspects of amino acid metabolism such as amino acid biosynthesis and nutrient transport [119,120]. By sensing and responding to the cellular availability of amino acids in the form of aminoacyl-tRNAs required for protein translation, T-box riboswitches play a major role in bacterial adaptation to rapidly changing nutritional environments. T-box-mediated gene regulation operates at both the levels of transcription (more common) and translation (less common) [47,110]. Specifically, stable binding of a cognate uncharged tRNA to the T-box riboswitch leads to transcriptional readthrough by stabilizing an otherwise weak antiterminator structure, which prevents premature termination of transcription by precluding the formation of an Rho-independent terminator hairpin. Similarly, for translational T-boxes, uncharged tRNA binding promotes translation initiation by formation of an anti-sequestrator helix that exposes the ribosome-binding site (or Shine–Dalgarno sequence). Due to a strong steric conflict between the tRNA aminoacyl group itself and the T-box antiterminator, aminoacylated tRNA cannot stabilize the nearly identical antiterminator or anti-sequestrator structure. This results in the formation of the competing transcriptional terminator hairpin or the Shine–Dalgarno sequence sequestrator helix, both of which shut off gene expression [50,51].

Several recently reported T-box–tRNA complex structures jointly paint a picture of multivalent, multimodal tRNA–mRNA interactions [47,48,51,114]. The T-box RNA wraps around the tRNA, making reinforced base-pairing interactions to its anticodon, stacking interactions to its elbow using its IDTM motif shared with RNase P, and base-pairing and base-stacking interactions with its 5′ and 3′ ends. Together, the flat termini of multiple adjacent dsRNA segments and stackable platforms (such as the tRNA elbow and T-box IDTM) concatenate to form an elongated spine, providing necessary stabilizing energy to turn on the T-box riboswitch.

### 6.4. SAM-Ixcc Riboswitch-tRNA Interactions

Recently, a novel tRNA-binding RNA element has been proposed in a group of SAM-I riboswitches [121]. SAM-I belongs to a diverse class of bacterial riboswitches that regulate gene expression in response to the intracellular availability of S-adenosylmethionine (SAM). The methionine biosynthesis operon in the phytopathogen *Xanthomonas campestris* pv. *campestris* (*Xcc*) is controlled through its 5′ UTR region encoding a SAM-I riboswitch at the level of translation initiation. This composite riboswitch is proposed to receive dual inputs of SAM binding to the canonical SAM-I portion of the RNA, as well as uncharged initiator tRNA binding to its 3′ region adjacent to the ribosome binding site [121]. Curiously, SAM-Ixcc RNA lacks the recognizable sequence or structural features used by RNase P and T-boxes, and exhibits no significant sequence complementarity to the tRNA. Therefore, it remains completely unknown how this RNA achieves tRNA recognition, and how it may sense and respond to tRNA aminoacylation.

## 7. Conclusions and Outlook

From the examples discussed above and other analyses, we derive the following conclusions, speculations, and projections towards an in-depth understanding of complex RNA–RNA interactions.

### 7.1. Frequent Facilitation of ssRNA–ssRNA Interactions by RNA Structure or Proteins

Although complementary ssRNA strands can in principle hybridize and interact with each other without assistance, biological ssRNA–ssRNA interactions in cells are generally facilitated by a third party. The facilitator can be adjacent structured RNA segments that help present the ssRNAs so that their intended base edges are exposed and pre-arranged for base-pairing formation with an incoming partner RNA. This is exemplified by the pre-organization of the tRNA anticodon trinucleotide structure [43]. By pre-organizing the interfaces, the system “prepays” a portion of the entropic cost of forming highly ordered contacts, thus lowering the energetic barrier of binding. Since RNA–RNA binding is generally driven by strong, favorable enthalpy from base pairing and base stacking formation, the total binding free energy becomes more favorable, resulting in higher affinity. In addition to pre-organizing the conformation of interaction motifs, RNA structural elements also provide the necessary overall topology and flexibility to facilitate multipartite interactions. Architectural motifs such as the K-turn control the global shape of RNA and, thus, the placements of key contact motifs to enable multiple contacts [122]. Flexible joints supply the conformational malleability necessary to engage dynamic RNA partners. This is exemplified by the C-loop motif in T-box Stem I, which pivots to allow the Stem I to track the flexing tRNA ligand about its 26•44 hinge [13].

Besides adjacent RNA structure, RNA-binding proteins frequently act to assist with RNA–RNA interactions in diverse ways. Protein binding to ssRNA prevents non-native, undesired self-structure formation such as self-hairpins and dimers, and extends the ssRNA to provide the topology closer to the bound form. In addition, RNA-binding proteins such as the L7Ae-superfamily can bind and stabilize RNA architectural elements such as K-turns, to support the intended global structure [58,123]. Furthermore, some RNA-binding proteins such as Hfq act as molecular chaperones that actively or passively remold the RNA conformations and bring the interaction motifs into proximity to catalyze binding [32,124].

### 7.2. Continued Underappreciation of Cross-Strand and End-to-End Stacking Interactions

ssRNA–ssRNA base-pairing produces dsRNA segments that are immediately flanked by adjacent ssRNA elements linking them to additional dsRNA segments. Although these flanking ssRNA residues are generally imagined as flexible tethers or linkers, they are in fact frequently observed as immobile elements that participate directly in the interactions. These ssRNA residues, especially purines, frequently act as helix caps that stack across the terminal base pair imparting necessary stabilizing energy, especially when the helical segment is short [125]. This is exemplified by the evolutionarily conserved and functionally critical A/G90 residue of the T-box Stem I, which stacks underneath the codon–anticodon duplex conferring required stability [13,47,126]. We anticipate observing many more such examples in upcoming RNA structures and suggest that when proposing secondary structure models conserved purines that immediately flank dsRNA segments be considered as potentially structured stabilizing elements as opposed to mere flexible linkers.

Another frequently underappreciated RNA–RNA interaction is between the termini of dsRNA segments. Topologically juxtaposed dsRNA helices are strongly attracted to each other to hide their hydrophobic base planes from the solvent. In both the HIV-1 PBS–tRNA and full-length T-box–tRNA complexes, we note a general tendency for the stems to concatenate via coaxial stacking to form long spines. This solution behavior likely gives rise to the formation of end-to-end stacked pseudo-infinite helices in the majority of RNA crystals [17,67]. In light of these observations, we anticipate a further emphasis of the key role of stacking interactions in shaping RNA structure and directing interactions.

### 7.3. Two Types of RNA Triplexes with Potentially Distinct Thermodynamic and Kinetic Properties and Attendant Functions

The ability to predictably and reliably perturb functionally important RNA structures or interactions using small, deliverable RNA devices is highly desirable for the design of effective RNA therapeutics. The recent structural elucidation of several ENE-polyA triplexes and discovery of the CCR5-miRNA triplexes suggest ample potential to leverage the controllable stability of RNA triplexes to modulate gene expression and cellular metabolic programs beyond RNA stability. For example, triplex elements can be exploited to conditionally control the formation or exposure of long dsRNA segments that trigger or dampen innate immune responses, or to regulate translation initiation. 

Considering the fact that most dsRNA-binding proteins access the shallow and wide minor groove, and the frequent occurrence of A-minor interactions, it seems there is the prospect to observe and create additional stable minor groove RNA triplexes beyond the BWYV triplex. Another notable difference is that minor groove triplexes seem to form with existing dsRNA segments, rather than by invading and reorganizing internal U-rich loops. Thus, major groove triplexes such as the ENE rely on substantial enthalpic gains to overcome large entropic costs of triplex formation, whereas minor groove triplexes such as the CCR5 and BWYV pseudoknots likely form with much reduced enthalpic drivers and entropic barriers. Conceptionally, this distinction may have a major impact on their thermodynamic and kinetic behavior and may further explain their different biological roles. It is possible that robust, high-barrier major groove triplexes are better suited for RNA protection against repeated probing by destructive enzymes, while more malleable minor groove triplexes function better as bistable regulators that can propagate an input signal of limited energetic signature (such as metabolite binding) to effect an RNA conformational change.

Taken together, our analyses highlight and dissect the symbiotic, reciprocal relationship between RNA structure and RNA–RNA interactions. The same set of operating principles and underlying forces enable ssRNA, dsRNA, triplexes, and other structural motifs to engage each other with affinity and selectivity, imparting diverse functionalities to noncoding RNAs as scaffolds, sensors, regulators, catalysts, and more.

## Figures and Tables

**Figure 1 ncrna-07-00081-f001:**
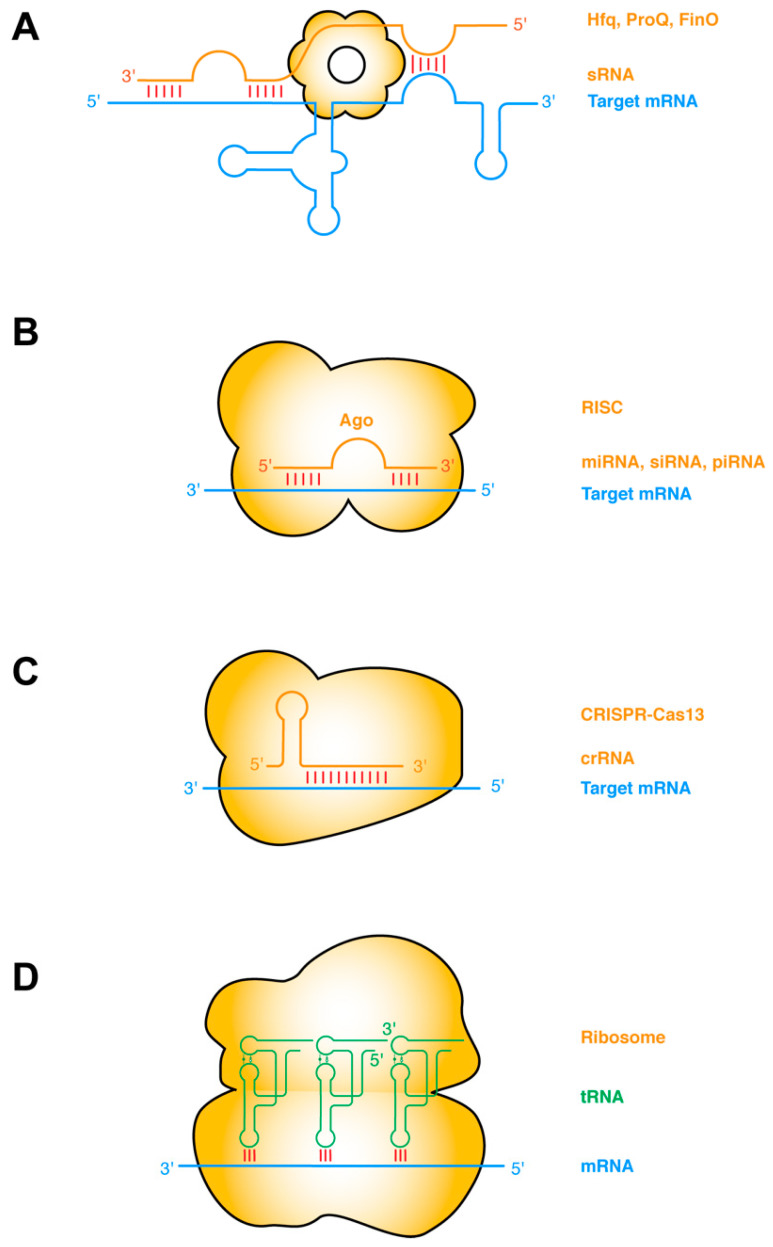
ssRNA–ssRNA interactions (**A**) Bacterial sRNAs form extensive base-pairing interactions with partially or wholly complementary segments of mRNAs, assisted by RNA chaperones such as Hfq, ProQ, FinO. sRNA binding remodels the mRNA structure to regulate its transcription, translation, stability, or decay, etc. (**B**) Eukaryotic small ssRNAs such as miRNAs, siRNAs, and piRNAs assemble with Ago proteins to form RNA-induced silencing complexes (RISC), and form perfect or imperfect base pairing with the target mRNA. Base pairing occurs first in the seed region and then in the supplementary pairing region. Such targeting leads to mRNA degradation or translation suppression. (**C**) RNA-targeting CRISPR-Cas13 binds the dsRNA duplex between the crispr RNA (crRNA) and the target mRNA, causing mRNA degradation and activating Cas13′s collateral RNase activity [34]. (**D**) Codon–anticodon base-pairing interactions between tRNA and mRNA are further stabilized by adjacent contacts from the rRNA and tRNA. Intermolecular base-pairing interactions are indicated by red sticks. Target mRNAs are shown in blue, sRNAs and other targeting or guiding RNAs in orange, and tRNAs in green. Proteins that facilitate the RNA interactions are shown as orange bodies.

**Figure 2 ncrna-07-00081-f002:**
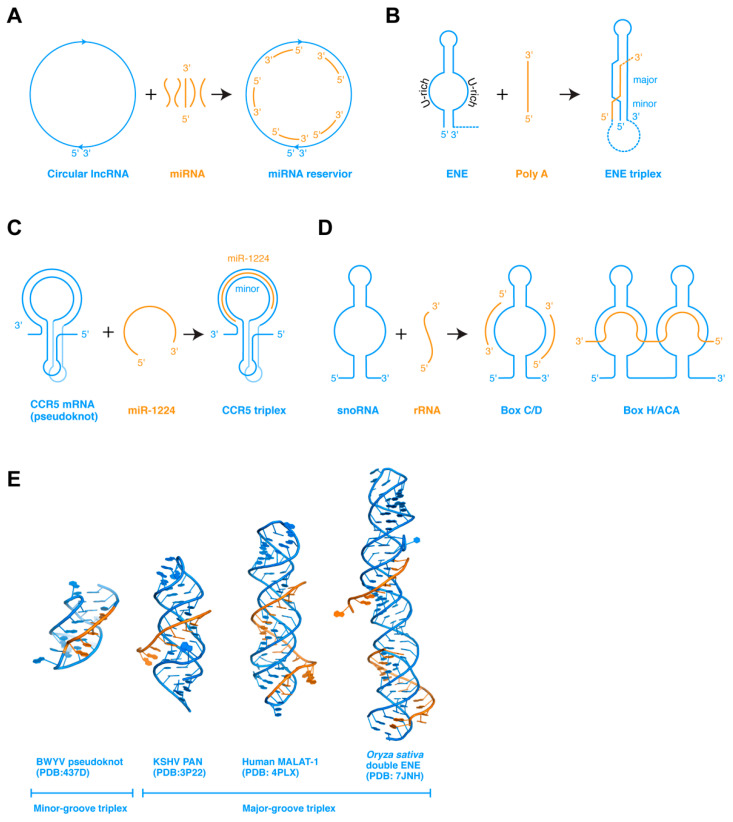
ssRNA interaction with structured RNAs. (**A**) Circular lncRNAs can associate with multiple miRNAs via base-pairing and store them until conditional release. (**B**) Expression and nuclear retention elements (ENEs) binds polyA ssRNAs forming RNA triplexes, leading to protection of the mRNA 3′ end from deadenylases and RNases. (**C**) The CCR5 pseudoknot is stabilized by miR-1224 forming an RNA triplex, promoting −1 programed ribosome frameshift. (**D**) SnoRNAs guide snoRNPs to their rRNA targets via base pairing and likely other tertiary interactions. (**E**) Crystal structures of the BWYV frameshifting pseudoknot and three ENE triplexes from KSHV PAN RNA, human MALAT-1 lncRNA, and rice double ENE RNA.

## Data Availability

Not applicable.

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
