# Peer review of "Cooperativity and Interdependency between RNA Structure and RNA–RNA Interactions"

_ncrna, 2021, doi:10.3390/ncrna7040081_

Round 1

Reviewer 1 Report

This comprehensive review on RNA-RNA interactions is very well written and gives very detailed information about RNA- RNA interactions and how the RNA structure influence the interaction or the other way around. All information is very useful for the researcher in the relevant field.

The manuscript is well organized but some of the writing should be edited to enhance clarity (see examples below).

  1. I would like to see the figures to be mentioned inside the paragraphs.
  2. The (a) in the legend should be changed to (A) which is shown in the figure.

  3. page 1,45, last paragraph "To address these questions ^ fundamental to the understanding"

Author Response

Point-by-point responses to reviewer comments:

We appreciate the positive comments and constructive suggestions from all three reviewers. Below we have responded to all the points raised.

Reviewer #1:

This comprehensive review on RNA-RNA interactions is very well written and gives very detailed information about RNA-RNA interactions and how the RNA structure influence the interaction or the other way around. All information is very useful for the researcher in the relevant field.

The manuscript is well organized but some of the writing should be edited to enhance clarity (see examples below).

  1. I would like to see the figures to be mentioned inside the paragraphs.

A:We thank the reviewer for her/his favorable comments and specific suggestions. We have added figure references to the text.

  1. The (a) in the legend should be changed to (A) which is shown in the figure.

A: Corrected as suggested.

  1. page 1,45, last paragraph "To address these questions ^fundamental to the understanding".

A: We simplified this to simply say:

 “To address these fundamental questions about the origin and characteristics of RNA-RNA interactions…”

Reviewer 2 Report

The manuscript By Skeparnias is a very nice review of RNA-RNA interactions.

What is particularly nice about the presentation is the build up from basic structural properties to very articulate examples with complex architecture and biological functions.

The reader can easily follow all examples even without having a prior specific knowledge of them as the basis are set and everything is built logically from them.

It is also remarkable the attention to link the arising of certain structures and interactions with energetic and entropic considerations, giving a physical understanding of the underlying process.

Author Response

We appreciate the positive comments and constructive suggestions from all three reviewers. Below we have responded to all the points raised.

We thank the reviewer for her/his favorable comments.  

Reviewer 3 Report

This review provides an overview of RNA:RNA interaction modalities, with an emphasis on the importance of RNA structure to bridge these interactions. Overall, this is a well-written manuscript, with a good introduction, clear figures illustrating the different classes of RNA:RNA contacts discussed by the authors—arranged in increasing order of complexity from simple reverse complementarity to quaternary contacts involving multiple interfaces between distinct RNA molecules, such as for RNase P or the T-box. The text is very informative, and will be of interest for any researcher in the broad field of RNA biology.

The conclusions are quite interesting and we agree with the authors of the importance of several key points, such as the assistance of RNA and proteins to “prepay” entropy for RNA:RNA contacts, and the underappreciated yet key role of the RNA bases in the vicinity of RNA:RNA contact segments.

Minor comments and suggestions:

(1) The choice of the word “symbiosis” in the title is quite strange when used to refer to RNA structure and interaction, given the scientific definition of this term. The authors probably meant to illustrate the key importance of RNA structure for RNA:RNA interactions, giving a similar emphasis on both aspects. I would suggest keeping only “interdependency”. (Alternatively: “Cooperation and interdependency”?)

(2) “The phosphate backbone of RNA is infrequently involved in specific RNA-RNA interactions due to their strongly repulsive negative charges”. This notion has been challenged with the discovery of a large amount of small RNA motifs that can involve the backbone, not only the sugar but also hydrogen bonds involving the phosphodiester linkages. Examples include ribosomal RNA and lysine bioswitches, for example; see the review by Ulyanov & James, 2010, New J Chem. which the authors may want to cite.

(3) When talking about G-quadruplexes (paragraph on line 131), it may be interesting to point out to the readers that these structures are actually more stable in RNA compared to DNA, and that they tend to fold slightly differently in RNA form, favoring parallel alignment (Joachimi et al, 2009, Bioorg Med Chem).

(4) The review should mention that it does not go into the detailed chemical structure of various RNA contact elements (examples: K-turns, subtypes of A-minor motifs, etc). To offset this, the authors may want to cite a couple chemistry-oriented reviews to fill this gap for the interested reader; (preferably one which presents the chemical structure for these).

(5) A potential improvement would be adding the annotation of the C-loop motif pivoting Stem I in the T-box riboswitch on Fig 3, which would make part of the text clearer for the reader (lines 512-514).

A few minor typos:

-line 23: “coordinately with” -> “in coordination with” would be clearer

-line 341: “underly” -> “underlie”

-the IDTM abbreviation is defined on line 409, but used previous on line 326

-the species name should be italicized on line 483-484, as: “Xanthomonas campestris pv. campestris” (except “pv.” standing for “pathovar”)

Author Response

We appreciate the positive comments and constructive suggestions from all three reviewers. Below we have responded to all the points raised.

Reviewer #3:
This review provides an overview of RNA:RNA interaction modalities, with an emphasis on the importance of RNA structure to bridge these interactions. Overall, this is a well-written manuscript, with a good introduction, clear figures illustrating the different classes of RNA:RNA contacts discussed by the authors—arranged in increasing order of complexity from simple reverse complementarity to quaternary contacts involving multiple interfaces between distinct RNA molecules, such as for RNase P or the T-box. The text is very informative, and will be of interest for any researcher in the broad field of RNA biology.

The conclusions are quite interesting and we agree with the authors of the importance of several key points, such as the assistance of RNA and proteins to “prepay” entropy for RNA:RNA contacts, and the underappreciated yet key role of the RNA bases in the vicinity of RNA:RNA contact segments.

A: We thank the reviewer for her/his thorough reading, favorable and insightful comments and detailed suggestions.

Minor comments and suggestions:

(1) The choice of the word “symbiosis” in the title is quite strange when used to refer to RNA structure and interaction, given the scientific definition of this term. The authors probably meant to illustrate the key importance of RNA structure for RNA:RNA interactions, giving a similar emphasis on both aspects. I would suggest keeping only “interdependency”. (Alternatively: “Cooperation and interdependency”?)

A: We agree with the reviewer and appreciate the suggestion. The revised title now states “cooperation and interdependency…”.

(2) “The phosphate backbone of RNA is infrequently involved in specific RNA-RNA interactions due to their strongly repulsive negative charges”. This notion has been challenged with the discovery of a large amount of small RNA motifs that can involve the backbone, not only the sugar but also hydrogen bonds involving the phosphodiester linkages. Examples include ribosomal RNA and lysine bioswitches, for example; see the review by Ulyanov & James, 2010, New J Chem. which the authors may want to cite.

A: We thank the reviewer for pointing out this important aspect that we overlooked. We have revised the text accordingly to highlight this recent discovery and added the citation. We now state:

“While the latter, ribose- and nucleobase-mediated contacts are the primary drivers of RNA-RNA interactions, long-range interactions that involve the phosphodiester back-bone are surprisingly common [4]. These generally involve OP1 or OP2 atoms as hydrogen bond acceptors and 2’-hydroxyl groups of an adjacently packed RNA segment as donors, forming the “ribose-phosphate zipper” motif [4].”

(3) When talking about G-quadruplexes (paragraph on line 131), it may be interesting to point out to the readers that these structures are actually more stable in RNA compared to DNA, and that they tend to fold slightly differently in RNA form, favoring parallel alignment (Joachimi et al, 2009, Bioorg Med Chem).

A: We have incorporated the reviewer comment on G-quadruplexes and the citations. We now state:

“These extensively paired, highly stacked structures exhibit diverse topological ar-rangements, are more stable in RNA compared to DNA, and tend to fold slightly differently, favoring the parallel alignment in RNA form [21, 22]”

(4) The review should mention that it does not go into the detailed chemical structure of various RNA contact elements (examples: K-turns, subtypes of A-minor motifs, etc). To offset this, the authors may want to cite a couple chemistry-oriented reviews to fill this gap for the interested reader; (preferably one which presents the chemical structure for these).

A: We have added this mention and citated relevant papers. We now state.

“More detailed analyses of the chemical structures of major RNA structural motifs such as K-turns, T-loops, A-minor motifs, and more, have been reviewed previously [25-29].”

(5) A potential improvement would be adding the annotation of the C-loop motif pivoting Stem I in the T-box riboswitch on Fig 3, which would make part of the text clearer for the reader (lines 512-514).

A: Added as suggested.

A few minor typos:

-line 23: “coordinately with” -> “in coordination with” would be clearer

-line 341: “underly” -> “underlie”

-the IDTM abbreviation is defined on line 409, but used previous on line 326

-the species name should be italicized on line 483-484, as: “Xanthomonas campestris pv. campestris” (except “pv.” standing for “pathovar”)

A:  All typos were corrected according to the reviewer’s suggestions.